# Regional Sustainability through Dispersal and Corridor Use of Asiatic Lion *Panthera leo persica* in the Eastern Greater Gir Landscape

Abhinav Mehta [1], Shrey Rakholia [1,2], Reuven Yosef [3,*], Alap Bhatt [1] and Shital Shukla [4]

1  The Geographic Information System (TGIS) Laboratory, Satellite, Ahmedabad 380015, GJ, India; mehta.abhinav01@gmail.com (A.M.); rakholias@gmail.com (S.R.); alapbhatt10@gmail.com (A.B.)
2  Bioinformatics Center, Forest Research Institute, Dehradun 248006, UK, India
3  Eilat Campus, Ben Gurion University of the Negev, P.O. Box 272, Eilat 881020, Israel
4  Department of Earth Sciences, Gujarat University, Ahmedabad 380009, GJ, India; shitalshukla25@yahoo.in
*  Correspondence: ryosef60@gmail.com

**Abstract:** Despite previous concerns regarding the survival of Asiatic Lions confined to the Gir Protected Area, their dispersal into surrounding landscapes has become a subject of considerable research and discussion. This study employs species distribution modeling, corridor analysis, and additional landscape assessment using satellite-based temperatures and Land Cover statistics to investigate this dispersal and identify potential corridors based on extensive field data. The results reveal the identification of a potential corridor from Gir Wildlife Sanctuary towards Velavadar Blackbuck National Park, indicating the expansion of the Asiatic Lion's range in the Eastern Greater Gir Landscape. These findings highlight the significance of resilience in Lion dispersal and corridor expansion, with implications for conservation and potential regional benefits, including ecosystem services and eco-tourism for sustainable development of the region.

**Keywords:** lion; metapopulation; species distribution modelling; felids; Big Cats; adaptation





## 1. Introduction

Once the most globally widespread mammal species, the lion (*Panthera leo* Linnaeus, 1758), one of the significant Big Cats, once roamed Eurasia and North America, including Alaska and Yukon, during the Pleistocene epoch [1]. Around 2.8–2.9 million years ago (mya), leopards (*P. pardus* Linnaeus, 1758) and lions diverged from a common ancestor [2,3]. However, evidence of post-divergence hybridization between ancient lineages of snow leopards (*P. uncia* Schreber, 1775) and lions is also apparent [4]. Then, a divergence between the Cave Lion (*P. spelaea* Goldfuss, 1810) and the modern lion occurred around 0.5 mya (de Manuel et al. 2020). Subsequently, another divergence occurred around 61,500 years ago (ya) between two main lineages: the North African/Asian and West and Central African lions of the modern lion, out of which two sub-species are extant currently, namely the African lion (*P. l. leo* Linnaeus, 1758) and the Southern African lion (*P. l. melanochaita* Smith, 1842). Later, around 51,000 ya, a split occurred between North African/Asian and West African lions. Lastly, the expansion of the North African lion started around 21,000 ya, resulting in the divergence of the Asian lion clade [5].

Historically, Asiatic lions (*P. l. persica* Meyer, 1826) ranged from North Africa, Eastern Europe, the Middle East, Northern Greece, Turkey, Iran, and Iraq to southwest Asia [6,7]. It is also notable that according to the revised taxonomic, the Asiatic Lion is considered an Asiatic subpopulation of *P. l. leo* [8]. However, for conservation purposes, legal aspects (As *P. l. persica* is mentioned under Schedule-1 with special protection from the Wildlife Protection Act 1972 of India [9]) and current usage by some authors recently [10,11], the use of sub-species name *P. l. persica* is continued in this study as well. The earliest known

documentation is the Old and New Testaments of the Holy Bible; in more than 100 different verses, the (Asiatic) lions are mentioned [12]. In the Indian sub-continent, their range extended from the Indo-Gangetic basin in the north to Odisha in the east, with their southern limits reaching the Narmada River [13]. However, they are confined to one isolated population of Asiatic lions in the Gir National Park (GNP) and Wildlife Sanctuary (WLS) and the Greater Gir landscape (GGL) of the Kathiawar Peninsula in Gujarat State, India. Over the last three decades, the lion population has increased by 137.32%. During the same period, its geographic area has expanded by 354.55% owing to the decades of effective conservation and protection measures by the state forest department and positive cultural attitudes of local communities towards lions and other wildlife [14,15].

Historically, the Asiatic lion population experienced two bottleneck events. The first is estimated to have occurred approximately 4279–1081 ya (median 2680 ya). This was due to the transformation of the Rann of Kutch into a bay or inlet around 4000–5000 ya, which effectively isolated the Saurashtra Peninsula and Gir Forest, making it inaccessible to mainland fauna until relatively recently. The second bottleneck event occurred more recently in the 19th century [16]. In the second bottleneck, the population of Asiatic lions declined dramatically, leaving around 50 individuals. This bottleneck was most likely the result of hunting, poaching, and habitat loss [17].

Previous research on Asiatic lions indicated high inbreeding and minimal genetic diversity [18]. Based on mitochondrial DNA (mtDNA) evidence collected from past bottlenecks and recent population restoration samples of lions in GNP, it is evident that the recent bottleneck has resulted in significantly lower genetic diversity in Asiatic lions compared to their African counterparts [19]. Therefore, the lack of genetic diversity and inbreeding can have worsening consequences for the population, as evidenced by a case study of a European captive population of Asiatic lions in a program initiated in the early 1990s in European zoos. By the late 2000s, this population experienced over 50% infant mortality, attributed to inbreeding. Consequently, the remaining population necessitated measures such as introducing individuals with different genotypes from India for breeding purposes [20].

Despite the bottlenecks, a genetic study found a relatively higher degree of polymorphism with more than 25% heterozygosity. This suggests the absence of intensive inbreeding, as previously reported. The low genetic variability in the actual Asiatic lion habitat, i.e., the Gir landscape, maybe a natural feature of the population rather than inbreeding, as the previous studies reporting lower genetic variation could be due to the usage of conventional experimental techniques, which had many limitations [21].

Also, a later study revealed positive indicators such as a healthy sperm count, a significant percentage of motile sperm, and a low occurrence of abnormal sperm—traits typically not observed in inbred animals. This suggests that inbreeding depression may not affect these lions, contrary to previous assertions. Additionally, elevated serum testosterone levels and strong fertilizing ability in semen suggest that Asiatic lions are not entirely subject to inbreeding [22]. Similarly, a study analyzing comparative sequence polymorphism among different lion groups as Asiatic lion (wild), Afro-Asiatic hybrid lions, and captive-bred Asiatic lions- revealed that wild Asiatic Lions exhibit similar polymorphism to captive-bred Afro-Asiatic hybrid lions and even more than captive-bred Asiatic ones. This suggests an apparent absence of intensive inbreeding and indicates the capability of surviving stress, as they represent a genetically healthy population [23]. However, when comparing heterozygosity levels among various Lion subpopulations in different datasets, the Asiatic lion showed lower heterozygosity for single nucleotide polymorphism (SNP) and microsatellites. It is worth noting that heterozygosity calculated from whole-genome data is underestimated, especially in samples with low coverage, similar to lion samples in Benin [24].

Unless the Asiatic lion population experiences severe inbreeding depression (extinction vortex), leading to a reduction in fitness and potentially facing extinction, it is not advisable to translocate African lion subpopulations into the Asiatic Lion habitat for genetic rescue. Instead, a metapopulation comprising interacting but spatially separated

populations in suitable areas surrounding the Gir Forest can be established. This approach can also protect against potential disease outbreaks [25]. Furthermore, previous attempts to establish a new Asiatic lion population or translocate them elsewhere far from the current habitat posed high risks of immunosuppression by pathogens and increased mortality rates due to arthropod-borne diseases, as noted in Etawah, Uttar Pradesh [26]. Attempts to reintroduce the Asiatic lion include one event where three individuals were introduced to Chandraprabha WLS in Uttar Pradesh in 1957. This effort initially had reproductive success, with the population growing to 11 individuals by 1965. However, the lions eventually vanished [27], and it is speculated that they had been shot or poisoned by the local populace [6].

Asiatic lion home range size is a dynamic interplay between habitat and individual characteristics [28]. Prey abundance, prey distribution, water availability, and factors like sex, age, pride size, and social status influence how much space a lion needs to roam [29]. Typically, males dominate in terms of territory size and overlapping pride ranges [30,31]. Studies in India in different areas showcase this variability: in the Gir PA, breeding females average 33 km$^2$ (95% kernel range size, 95% MCP was 35 km$^2$, and 100% MCP range was 49 km$^2$; [32]), while lions in coastal habitats require almost five times that range (average 171.8 km$^2$; [33]). Gender also plays a role, with males in coastal areas averaging 85 $\pm$ 54 km$^2$ compared to females' 48.2 $\pm$ 10.6 km$^2$ and 35 $\pm$ 7 km$^2$ [30]. Even within the Eastern Gir landscape, geographical ranges differ significantly, with lions requiring 621 km$^2$ compared to their 243 km$^2$ counterparts in the Girnar wildlife sanctuary [34]. Similarly, the female home range was 129 km$^2$ compared to 170 km$^2$ in the Girnar WLS. To delve deeper into movement patterns, Ram et al. [29] radio-collared 22 individuals, revealing annual home ranges from 15 to 415 km$^2$. This research highlights the diverse factors shaping how Asiatic lions navigate their environment.

Asiatic lions exhibit unique behaviors shaped by their environment and prey dynamics [31]. Unlike African lions that have migrating prey and shifting territories [35], female Asiatic lions maintain stable pride territories due to evenly distributed, non-migratory prey like chital (*Axis axis* Erxleben, 1777) and sambar (*Rusa unicolor* Kerr, 1792). This stability allows female home ranges to overlap multiple male ranges, and these females also show a unique behavior called multi-male mating. This multi-mating strategy reduces cub infanticide by males. It potentially diversifies offspring lineages unseen in African lions. However, female African lions have a maternal grouping strategy where multiple females benefit from grouping, which protects the cubs from infanticidal males [36,37]. Males, in turn, adapt by forming coalitions, sharing resources, and mating opportunities despite asymmetries. Though prey is smaller and chances for simultaneous mating are lower, this behavior increases reproductive success for subordinates compared to solitary males. Consequently, multiple male home ranges can overlap these female territories, leading to females' multi-male mating strategy unique to Asiatic Lions [38].

Prey availability heavily influences the distribution of Asiatic lions within protected areas [39]. Chital and sambar deer comprise over 50% of their diet, followed by a smaller portion of domesticated livestock like cattle and buffalo (around 20%). Thanks to its even distribution and non-migratory nature, this stable prey base allows female lions to maintain stable pride territories [40].

Life outside protected areas presents a different scenario for lions. Here, wild prey, particularly chital and sambar, are scarce. Consequently, lions adapt by relying more on domesticated livestock (around 42%). Nilgai (*Boselaphus tragocamelus* Pallas, 1766) and wild boar (*Sus scrofa* Linnaeus, 1758) emerge as their primary wild prey sources, comprising roughly 24% and 15% of their diet. This shift in prey dynamics underscores the importance of understanding the lions' habitat distribution and the variations in prey availability across different landscapes [41].

The distribution of Asiatic lions is not only influenced by prey dynamics within protected areas but also by the surrounding landscape. In the last two decades, lion populations have spilled over into the eastern landscape (Ambardi-Savarkundla belt in

Amreli and Satrunji-Hippavadli belt in Bhavnagar district). This eastward expansion is primarily driven by the availability of wild prey like nilgai and livestock, as westward movement is restricted by poor habitat quality due to intensive agriculture and human settlements [42]. Also, the latest comprehensive assessment shows a total of 674 lions, including 344 lions inside the Gir PA and 330 outside the Gir PA, with at least 250 lions ranging eastward of the Gir PA [10]. Our study focuses explicitly on the Asiatic lion's expanding range outside Gir PA, known as the Eastern Greater Gir Landscape (EGGL). We employed advanced methodologies like Maxent-species distribution modeling, Linkage Mapper-linkage pathways corridor analysis, and satellite datasets to analyze this critical habitat and identify potential corridors for lion movement.

## 2. Materials and Methods

### 2.1. Study Area

Located in the Kathiawar (Saurashtra) region east of the Gir Protected Area, aka Gir PA (with a total area of 1410 km² comprising both Gir NP and Gir Wildlife Sanctuary), the EGGL stretches across subdistricts (Talukas) primarily within Amreli and Bhavnagar Districts of Gujarat, India (Figure 1). This landscape facilitates consistent animal movement between these districts, spanning geographical coordinates from approximately 20°45′ to 22°7′ N latitude and 71°5′ to 72°22′ E longitude [43]. According to the Köppen-Geiger climate classification map, the EGGL has a hot Semi-arid climate (Bsh) characterized by hot, dry summers and mild winters [44].

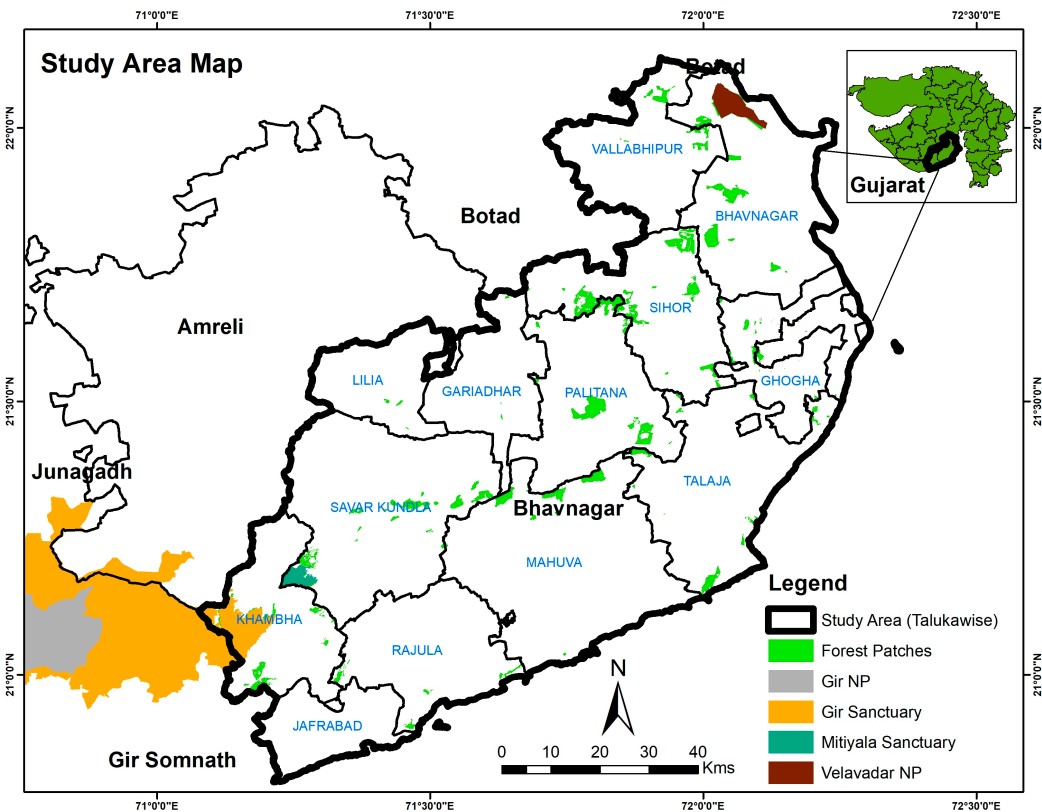

**Figure 1.** Study area map of Eastern Greater Gir Landscape and the subdistricts (talukas) it encompasses.

Our study utilizes species distribution modeling (SDM), specifically the MaxEnt model, to gain insight into the eastward expansion of Asiatic lions beyond the Gir PA. This approach builds upon ecological and biogeographical principles linking species distribution with their physical environment [45,46].

To develop the SDM, we employed a diverse set of 25 environmental layers. These layers encompassed:

- Six land cover layers specific to the study area (2019 data), including distance to roads, water bodies, elevation, slope, and aspect.
- Nineteen bioclimatic layers were retrieved from the WorldClim website at a resolution of 1 km$^2$ (TIFF format).

This comprehensive data collection ensures a robust analysis. Furthermore, we leveraged over 4700 Asiatic lion occurrence points for 2018–2019. These points originated from various sources, including direct field sightings, forest department census data, regular "full moon" surveys, and livestock depredation datasets from Amreli and Bhavnagar forest divisions.

Even after spatial thinning, we retained a significant sample size of 3087 occurrences for the MaxEnt model. This aligns with established recommendations emphasizing spatially distributed occurrence points and duplicate removal [47].

We imported bioclimatic and environmental layers and occurrence data into QGIS using the Point Sampling Plugin to integrate these diverse data sources. Subsequently, we carried out the MaxEnt-recommended variable selection method, followed by correlation analysis to identify relationships among variables. This analysis employed Pearson's correlation method, with coefficients ranging from 0 to 1. We retained variables with correlation coefficients exceeding 0.7. This rigorous selection process led to the inclusion of 13 out of the 25 bioclimatic variables in the final MaxEnt model (Table 1).

**Table 1.** Table showing estimates of the relative contributions of the environmental variables to the Maxent model.

| # | Bioclimatic/Environmental Variable | Percentage Contribution | Permutation Importance |
|---|---|---|---|
| 1 | Bio2 (Mean Diurnal Range (Mean of monthly (max temp − min temp))) | 59.6 | 55.2 |
| 2 | Land Cover [1] | 15.6 | 6.1 |
| 3 | Elevation | 8.5 | 5.9 |
| 4 | Bio1 (Annual Mean Temperature) | 6 | 5.7 |
| 5 | Distance to Road | 2.3 | 4.6 |
| 6 | Bio14 (Precipitation of Driest Month) | 1.6 | 3.3 |
| 7 | Bio18 (Precipitation of Warmest Quarter) | 1.6 | 7.4 |
| 8 | Bio3 (Isothermality) | 1.2 | 3.5 |
| 9 | Bio12 (Annual Precipitation) | 1 | 2 |
| 10 | Bio15 (Precipitation Seasonality (Coefficient of Variation)) | 0.9 | 1.8 |
| 11 | Distance to Waterbodies | 0.8 | 2.7 |
| 12 | Aspect | 0.7 | 1.2 |
| 13 | Slope | 0.3 | 0.7 |

[1] The provisional name of Land Cover used in model was lu1000.

MaxEnt model outputs included Jackknife, Response curves, and raster outputs in .asc (ASCII) file formats. These outputs provided insights into habitat suitability, which are detailed in the Results section.

To understand how Asiatic lions navigate the Eastern Greater Gir Landscape (EGGL), we employed the Linkage Mapper-Linkage Pathway Tool, an ArcGIS toolbox specifically designed to map connections between critical habitats. This tool assesses the potential for movement across different pathways by considering factors that could aid or hinder animal travel. The analysis relies on two key datasets:

- Core areas: Represented by designated "Notified Forests" (government-protected forest categories like Reserved, Protected, and Unclassified), areas identified based on both field observations and insights from the SDM results.
- Land cover: This data creates a "resistance layer" by assigning weights to various land cover types. These weights reflect the relative difficulty or ease of movement for lions across different terrains.

We assigned each land cover class's weight (rank) such that a higher weight (higher resistance) results in a higher cost for lion movement. Built-up areas, water bodies, and streams, with weights of 13, 12, and 11, are likely to pose higher barriers or risks to lion movement. Sandy beaches (10) and salt pans (9) follow close behind, suggesting they also

hinder the movement of the lions. As weights decrease, indicating reduced barriers or costs to lion movement, Salt-affected land (8), barren lands (7), and mangroves (6) are assigned decreasing weights. Fallowland and agricultural farmland, with weights of 5 and 4, offer slightly easier passage for lions, while vegetation areas, grasslands, and forest areas, i.e., notified forests, with weights of 3, 2, and 1, are probably the most suitable for lion movements due to their lower obstacles or risks.

By analyzing this network of core areas and their connections, considering distance and landscape resistance, our study identifies the least-cost paths between these critical habitats. These paths represent the most likely routes lions might use to disperse and connect across the EGGL landscape [48]. This approach provides valuable insights into the landscape connectivity patterns crucial for lion conservation strategies.

### 2.2. Additional Satellite-Based Land Surface Temperatures (LST) Analysis

Building upon our species distribution modeling (SDM) results, we further investigated the significance of temperature variations for Asiatic lions. Employing Google Earth Engine (GEE) scripts, we analyzed Land Surface Temperatures (LSTs) derived from the Thermal Infrared bands of Landsat 8 for 2018–2019. This analysis focused on mean LSTs across different land cover classes.

The land cover data was specifically prepared for the 2019 post-monsoon season (November-December), considering its relevance to the SDM findings. Based on the findings, we chose three key variables for comparison:

- Bio1 (Annual Mean Temperature)
- Bio2 (Mean Diurnal Range)
- Land Cover (LULC)

This selection was motivated using relatively high-resolution LST data at approximately 100 m to provide detailed insights.

For analysis purposes, we grouped grassland, scrubland, and sparse to moderately dense vegetation outside notified forests into a category named "Natural Vegetation/Plantations" (also encompassing potential mango plantations). This simplification facilitated statistical analysis conducted using R version 4.3.0.

## 3. Results

### 3.1. Species Distribution Model Results

The area under the curve (AUC) is a graph of sensitivity (how well the data correctly predicts presence) and specificity (how well the data correctly predicts absences). The AUC values vary between 0.5 and 1, with the former indicating a model that is no better than chance and the latter providing increasingly better discrimination as it approaches 1, i.e., 0.5–0.6 = no discrimination; 0.6–0.7 = discrimination; 0.7–0.8 = acceptable; 0.8–0.9 = excellent; 0.9–1.0 = exceptional [49]. The SDM MaxEnt model for Asiatic Lion showed an AUC of 0.766 (SD = 0.014), which represents an acceptable performance of the model given the large sample size (Figure 2).

Bio2 (Mean Diurnal Range) and Land Cover contributed 59.6% and 15.6%, respectively and were the two largest contributors to the MaxEnt model, and therefore of very high importance for the analysis (Table 1).

The particularly suitable habitat areas were the core areas (forest patches aka Notified Forests) starting from the west of the study area towards the east from Khambha to Palitana sub-districts (Figures 1 and 3). Therefore, we selected these notified forests as core areas for the following corridor analysis, described in detail in the following sub-section. In addition, some of the coastal habitats to the south were also moderate to highly suitable.

Additionally, the results of the Jackknife test showed that Bio1 (mean annual temperature) was also an important variable, providing the most helpful information on its own. In contrast, the land cover variable (lu1000) showed decreasing increments when omitted, suggesting that most of the unique information was missing in other variables (Figure 4).

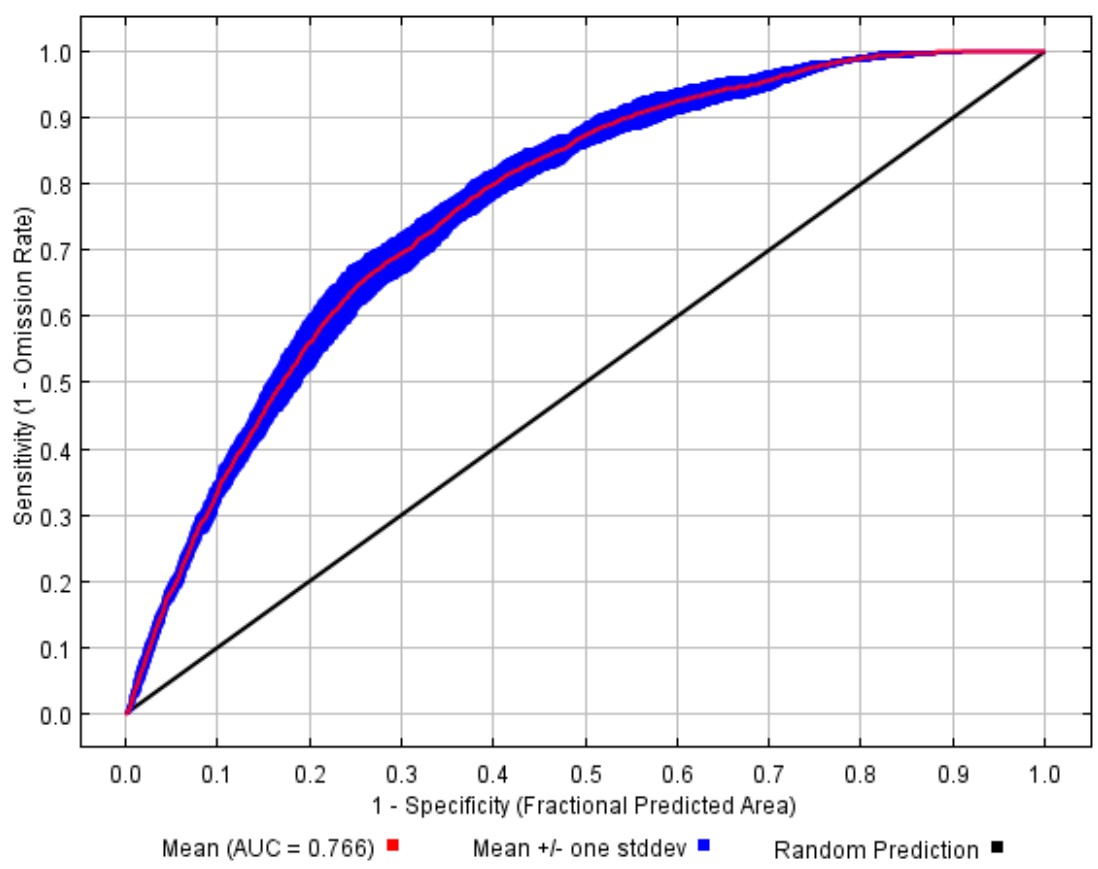

**Figure 2.** Receiver operating characteristic (ROC) curve showing average sensitivity vs. specificity for the Asiatic lion *P. leo persica*.

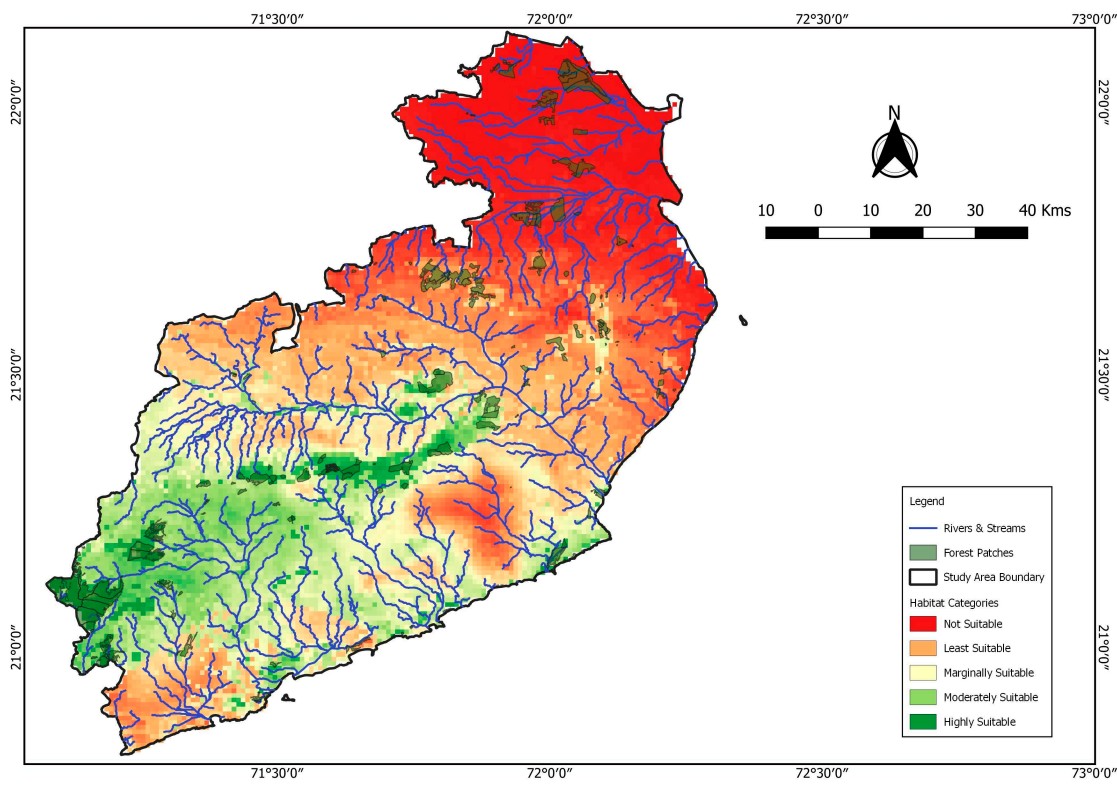

**Figure 3.** Habitat Suitability Map of Asiatic Lion based on the species distribution modeling in MaxEnt.

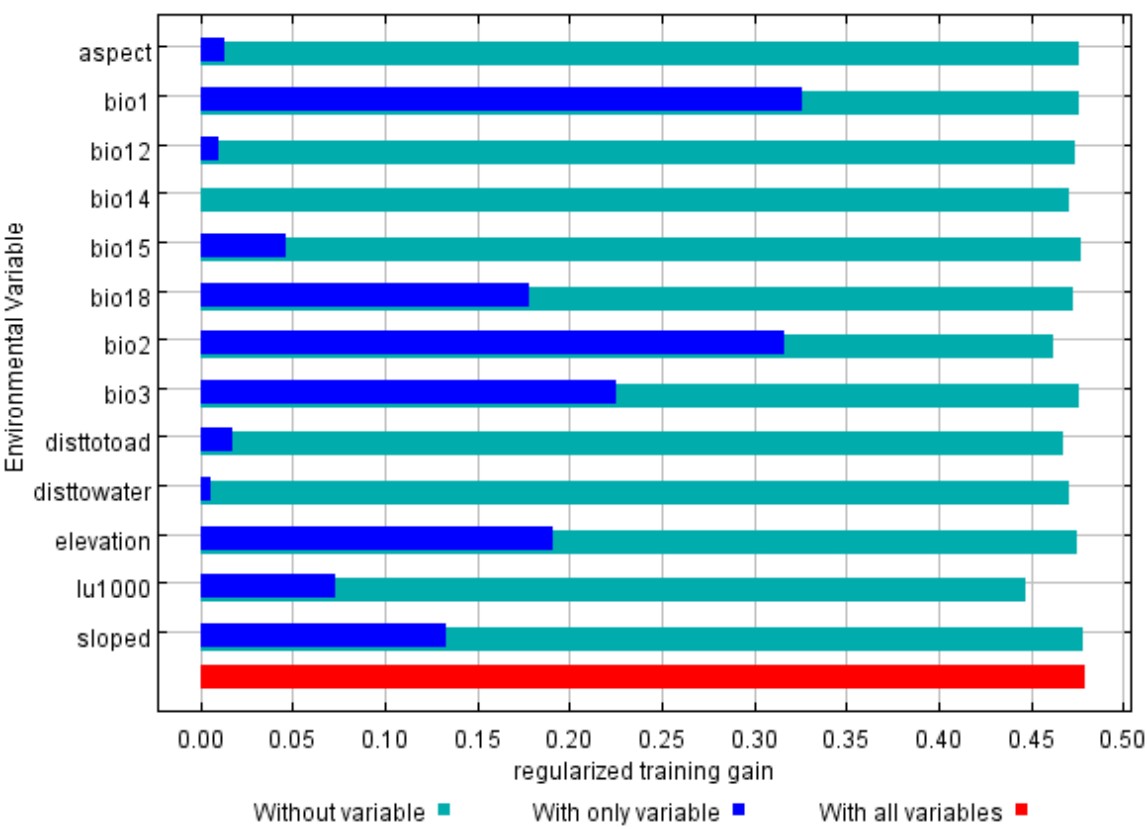

**Figure 4.** Jackknife plot for the test of variable importance. Darker blue shades show the gain in isolation from other variables.

Therefore, we selected the variables Bio1, Bio2, and Land Cover for further analysis using LSTs to represent Bio1 and Bio2 at relatively higher resolution.

### 3.2. Corridor Analysis Results

The results of the corridor analysis have three outputs: (a) corridor linkages, (b) multiple paths, and (c) prominent corridor patches. The centrality links connecting all core forest patches (notified forests) show how the core regions are interconnected and could serve as alternative habitats for the lion population (Figure 5). The central patch covering the core forest patches (notified forests) has several connections from west to east, but alternative connections exist from the southern coastal routes (Figure 5). However, based on the estimation of cost-raster least-cost paths, the coastal routes have few multi-link- routes running west to east from Gir Sanctuary to Velavadar (Blackbuck) NP through the Notified Forest patches in the middle (Figure 6).

The main lengths of the corridor sections start from a single section of the corridor, namely Gir (1) to Jesar, followed by two alternative corridor paths, namely Jesar-Piparala (2a) and Jesar-Thala (2b), both occupied by lions, followed by two other alternative corridor paths, Piparala-Velavadar NP (3a) and Thala-Velavadar NP (3b; Table 2, Figure 7).

### 3.3. Land Surface Temperature-Land Cover comparison

Bio2 response curves show that the occurrence prediction or the habitat suitability is higher in lower diurnal ranges, i.e., 9.5 °C to 10 °C (Figure 8). This implies that habitat suitability is higher in lower diurnal temperature ranges. Since Bio2 is the highest contributing variable in the model (Table 1), along with Bio1, which has higher importance as an isolated variable, it was essential to analyze this further with observed satellite datasets, i.e., Landsat 8 LSTs resolution is finer compared to the MaxEnt's bioclimatic variable datasets.

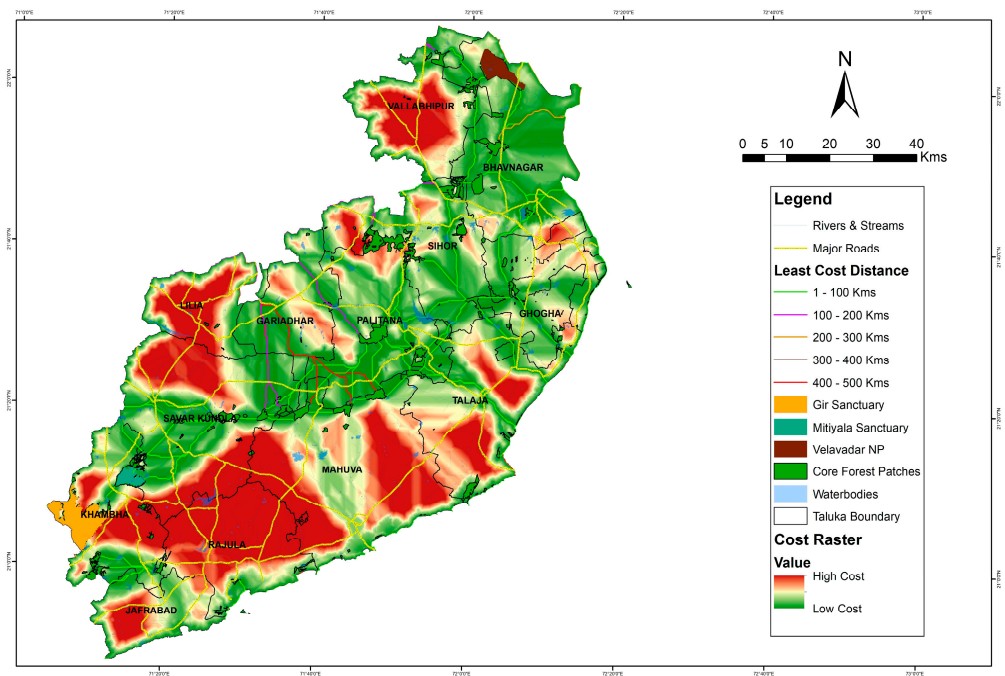

**Figure 5.** The Corridor Centrality Linkages map showing multiple possible linkages.

**Figure 6.** The Potential Corridor least-cost paths map showing all potential paths.

**Table 2.** The potential corridors as per the linkage pathway analysis.

| # | Corridor | Sub-Districts (Talukas) | District(s) | Distance |
|---|----------|-------------------------|-------------|----------|
| 1 | Gir-Mitiyala-Vijaynagar-NaniVadal-Kantrodi-Jesar | Khambha, Savarkundla& Mahuva | Amreli & Bhavnagar | 55 Kms |
| 2a | Jesar-Palitana-Anida-Piparala OR | Mahuva -Palitana-Sihor | Bhavnagar | 49 Kms |
| 2b | Jesar-Beda-Karmadiya-Ratanpar-Sanjnasar-Nani Rajsthali-Thala | Mahuva -Palitana-Sihor | Bhavnagar | 48 Kms |
| 3a | Piparala-Ghanghali-Nesda-Savainagar-NavaMadhiya-Rajgadh-Mevasa-Velavadar NP OR | Sihor-Vallabhipur & Bhavnagar | Bhavnagar | 45 Kms |
| 3b | Thala-Ghanghali-Nesda-Savainagar-NavaMadhiya-Rajgadh-Mevasa-Velavadar NP | Sihor-Vallabhipur & Bhavnagar | Bhavnagar | 56 Kms |
| **Total Length of Potential Corridor** | | | | **149–159 Kms** |

Note: There are three major patch lengths (paths) for the potential corridors with two alternative patch lengths (2a and 2b; 3a and 3b).

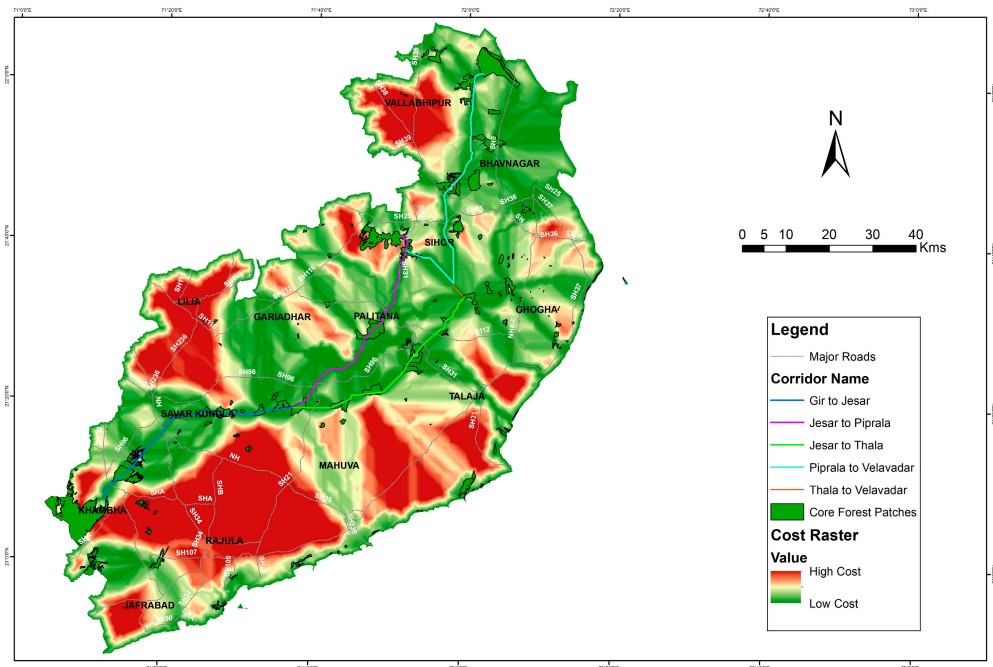

**Figure 7.** Main Potential Corridor Map showing major corridors.

Based on the ridge plot analysis, we concluded that the occurrences of lions are higher in agricultural areas with higher mean LST values (43.6 °C in fallow land and 43.3 °C in cropland) above the mean LST values. However, lions occurred in all habitats, including aquatic environments (water bodies can include riparian habitats, ponds, wetlands, etc.) and coastal habitats such as salt-affected land, mangroves, and coastal vegetation. Nevertheless, lion occurrences are extensive in relatively temperate habitats such as Notified forests and natural vegetation, with mean LST values of 41.4 °C and 42.18 °C, respectively (Figure 9).

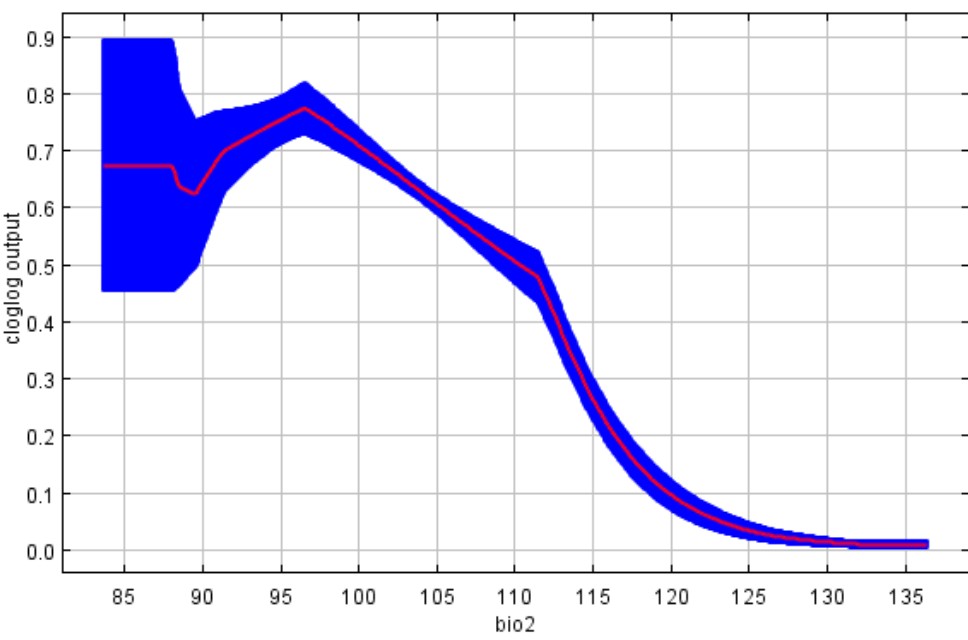

**Figure 8.** Marginal response curve of bio2 Mean Diurnal Temperature Range with a predicted probability of presence (cloglog output; in blue). Note: The Bio2 values (in °C) are scaled by a factor of 10.

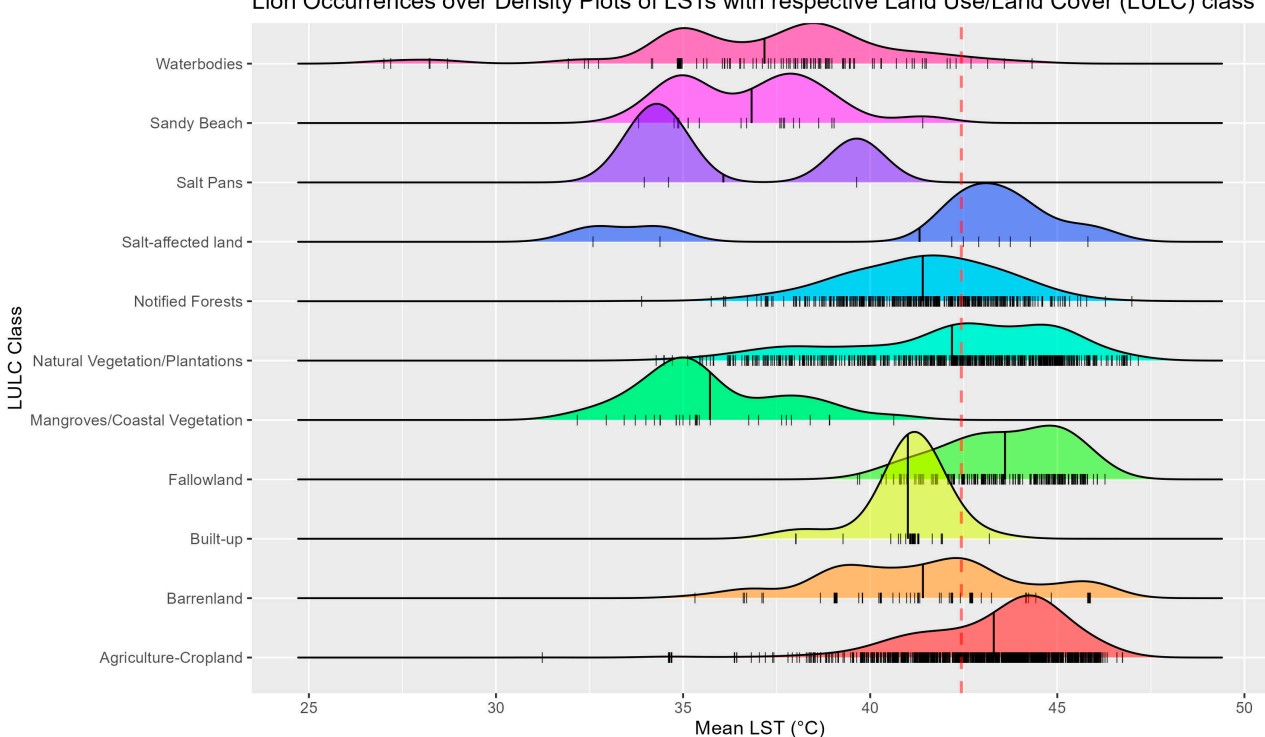

**Figure 9.** Ridge Plot of Mean LSTs (2018–2019) with respective Land Cover class in lion occurrence locations. Note: the Black jitter data points are shown as '|', LST ranges (in °C) are shown in the density plots in each land cover class and Red line shows overall mean (42.4 ± 2.7 °C).

## 4. Discussion

The Asiatic lion prefers a low mean diurnal range (bioclimatic variable) similar to what is found in this study, which determines its distribution in the EGGL [43]. As mentioned earlier, chital and sambar constitute the primary prey species in the Gir PA. However, in the



Bhavnagar district, the vidis (open thorn scrub or tropical savanna grasslands) are mainly inhabited by nilgai, chinkara (*Gazella bennettii* Sykes, 1831), and wild boar. At the same time, sambar, chital, and blackbuck (*Antilope cervicapra* Linnaeus, 1758) are negligible [29]. In our LULC categories, scrublands and grasslands fall under the natural vegetation class. Additionally, deer species had higher densities in forests than in grasslands and coastal habitats, as the Asiatic lion prefers larger prey such as sambar; this has reduced population density equilibrium in Gir PA in recent decades [50]. This further suggests that the mean diurnal bioclimatic variable significantly influences the distribution of the Asiatic lion, highlighting its preference towards areas with dense vegetation, such as forests over grasslands and coastal forests [31].

The analysis reveals a clear preference of Asiatic lions for dense vegetation habitats, particularly moist-mixed and Teak-Acacia-Zizyphus forests, constituting roughly 65% of their habitat use, with less than <10% of the thorn habitat. In contrast, open habitats like thorn forests are less frequented. This preference aligns with the observed habitat use of their primary prey, chital and sambar. Conversely, nilgai exhibits a higher preference for thorn forests, with around 20% of their habitat use occurring there. This suggests a greater reliance on nilgai as prey in these less dense thorn forests [51].

Interestingly, our results show that bioclimatic variables like mean diurnal range play a significant role in species distribution. By integrating this information with observed habitat use patterns of both lions and their prey, we understand why this variable is essential. The preference for denser vegetation with lower diurnal temperature variations might create microclimates favorable for both lions and their preferred prey, contributing to their higher occurrence in those habitats. Concerns have been voiced regarding the survival of Asiatic lions outside the Gir PA (including GNP, WLS, and other Protected Forests and Reserve Forests) previously [52]. These concerns often focus on potential declines in connecting corridors for lion movement. However, our study presents compelling evidence that challenges these concerns. Based on our findings, lions outside the Gir PA exhibit robust movement and utilize multiple connecting corridors, suggesting a healthy and expanding population as evidenced by occurrence records. This is further supported by identifying the eastern dispersal corridor as part of the established EGGL, an area recognized for its high concentration of lion predation hotspots [53]. This corridor stretches from the East of Gir Wildlife Sanctuary through Savarkundla and Palitana, reaching Velavadar NP in Bhavnagar.

Furthermore, our analysis reveals increasing landscape connectivity, characterized by a growing network of interconnected forest and vegetation patches. This larger, integrated habitat facilitates wildlife movement significantly [54]. The increase in post-monsoon vegetation patches acts as a buffer, promoting lion movement from protected forests to surrounding areas. This positive trend might be associated with increasing rainfall observed in the region [55]. Studies analyzing 146 years of precipitation data confirm an upward trend in annual southwest monsoon rainfall across Saurashtra [56].

Additionally, compared to other Saurashtra districts, Amreli and Bhavnagar exhibit lesser variability in monthly monsoon rainfall, suggesting a lower risk of extreme or highly variable climatic events even with rising monsoon trends [57]. This combined evidence effectively challenges previous concerns about dwindling connectivity and declining lion populations outside the Gir PA. Instead, our research shows a thriving lion population with many opportunities for movement and expansion because of robust connecting corridors, an increasingly interconnected landscape, and favorable climatic conditions. This is similar to Kittle et al. [28], who found that landscape influenced space use and the seasonal vegetation that influenced prey vulnerability to predation.

Our study reveals a strong correlation between dense vegetation patches and increased lion presence. This finding aligns with the findings of Vijayan and Pati [58,59], who identified a trend of shifting crop patterns, with an increase in mango (*Mangifera indica* L.) and sugarcane (*Saccharum officinarum* L.) plantations, coinciding with enhanced lion movement in those areas. The potential explanation lies in the cover and prey opportunities offered by

these denser vegetated areas. Within the EGGL, our analysis demonstrates that lions utilize a variety of land cover types. Their presence extends beyond notified forests to include areas like natural vegetation/plantations, barren land, fallow land, and even agricultural land (excluding built-up areas). This underlines the Asiatic lion's adaptability and ability to navigate diverse landscapes for suitable habitats. Our findings resonate with existing research by Ram et al. (2022) [33], who identified similar high (>65%) probabilities of lion occurrence in forested areas, dense scrubland, mangroves, and sandy areas. Interestingly, Nowak (2013) [60] notes the presence of other felids like leopards and tigers (*P. tigris* Linnaeus, 1758) in mangroves, while African lions are known to also utilize inter-tidal zones for foraging [61]. These comparisons highlight the unique habitat preferences of Asiatic lions within the broader context of felid ecology. In addition, this analysis reveals lion occurrences within areas experiencing higher LSTs compared to their natural forested habitats like the Gir Protected Area. This apparent presence in areas with warmer temperatures might seem counterintuitive. However, it could also be an adaptive mechanism, allowing lions to exploit resources in non-forested landscapes like agricultural croplands and fallow lands. While our model identifies mean diurnal range as a significant contributor to lion distribution, with a higher probability associated with lower temperature variations, the observed presence of lions in higher LST regimes suggests a complex interplay between climatic preferences and resource availability [62]. In simpler terms, lions might tolerate higher temperatures by accessing valuable resources offered by specific land cover types. Using LST data from Landsat 8 provides higher resolution information than the Bioclim data. This higher resolution analysis provides more insights into lion distribution that might be missed using broader-scale analyses, as evidenced by the importance of annual mean temperature (Bio 1) when considering individual variables in our Jackknife analysis. Our findings resonate with broader ecological patterns observed globally. Similar studies like Gallou et al., 2023 [63] have identified a negative correlation between species range size and diurnal temperature range. This suggests that some species, like lions in our study, might adjust their range distribution to access essential resources even in less preferred temperature regimes.

Our study's corridor analysis extends beyond simply identifying potential lion movement pathways. The almost 150 km dispersal corridor connecting the Gir Protected Area to Blackbuck National Park (Velavadar NP) through notified forests and natural vegetation buffers opens doors for broader discussions and future possibilities. This long-distance dispersal highlights the need for coordinated management strategies across diverse landscapes, encompassing protected areas, corridors, and surrounding human-dominated areas [54,64]. Balancing lion conservation with human activities and addressing potential conflicts will be crucial [65]. The expanding lion range presents exciting regional eco-tourism opportunities. Responsible and well-managed tourism can contribute to sustainable local economic development while fostering public appreciation for lion conservation. Lion presence could boost ecosystem services within the region. The observed dispersal pattern suggests the potential for establishing a lion metapopulation without human intervention. This naturally dispersing and adaptable population could enhance the long-term viability of the species [66].

Our findings shed new light on lion dispersal and corridor dynamics from environmental, bioclimatic, geographical, and behavioral perspectives.

## 5. Conclusions

Based on our findings, Lion's dispersal and expansion in terms of corridors is significant, and this has multiple implications and opportunities in conservation and potential benefits to this region both in terms of ecosystem services and in terms of development of this region in terms of eco-tourism [67].

Furthermore, this trajectory of dispersal may lead towards a healthy natural metapopulation in the future without the need for translocation, as metapopulation is sufficient for management during outbreaks rather than establishing a separate population by translo-

cating as it has higher risks. In addition, these findings suggest that the Asiatic Lion is a highly adaptive subspecies occurring in diverse regional habitats regarding foraging, mating, and dispersal behavior. Therefore, future studies should focus on this subpopulation's resilience and adaptive mechanisms. These conclusions underscore the importance of considering lion dispersal patterns, corridor dynamics, and sustainable management practices to ensure the long-term survival and well-being of Asiatic Lions in the Eastern Greater Gir Landscape.

**Author Contributions:** Conceptualization, A.M., S.R. and R.Y.; methodology A.M. and S.R.; software, A.M. and S.R; validation, A.M., S.R., R.Y., A.B. and S.S.; formal analysis, A.M., S.R., R.Y. and A.B.; investigation, A.M., S.R., R.Y., A.B. and S.S.; resources, A.M., S.R. and R.Y.; data curation, A.M. and S.R.; writing—original draft preparation, A.M., S.R., R.Y. and A.B.; writing—review and editing, S.R., R.Y. and A.B.; visualization, A.M. and S.R; supervision, R.Y. and S.S.; project administration, S.R. All authors have read and agreed to the published version of the manuscript.

**Funding:** This research received no external funding.

**Informed Consent Statement:** Not applicable.

**Data Availability Statement:** The data are not publicly available owing to conservation concerns and restrictions of the Gujarat Forest Department.

**Acknowledgments:** We express our gratitude to the Chief Wildlife Warden, Gujarat State Forest Department, for permission and assistance in data collection.

**Conflicts of Interest:** The authors declare no conflicts of interest.

## Abbreviations

| | |
|---|---|
| ASCII | American Standard Code for Information Interchange |
| AUC | Area Under Curve |
| EGGL | Eastern Greater Gir Landscape |
| GEE | Google Earth Engine |
| GGL | Greater Gir Landscape |
| GNP | Gir National Park |
| LST | Land Surface Temperature |
| LULC | Land use-Land Cover |
| MCP | Minimum Convex Polygon |
| mtDNA | Mitochondrial Deoxyribonucleic acid |
| NP | National Park |
| PA | Protected Area |
| ROC | Receiver Operating Characteristic |
| SDM | Species distribution Modelling |
| SNP | Single Nucleotide Polymorphism |
| TIFF | Tag Image File Format |
| WLS | Wildlife Sanctuary |

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
