# Peer review of "Regional Sustainability through Dispersal and Corridor Use of Asiatic Lion Panthera leo persica in the Eastern Greater Gir Landscape"

_sustainability, doi:10.3390/su16062554_

Round 1
Reviewer 1 Report
Comments and Suggestions for Authors
The present study is a formally, methodically and factually well-crafted contribution to the knowledge of the conservation status of Asiatic lions and an important contribution to the practice and theory of their conservation. I strongly recommend its acceptance. I have no major reservations. Just a few minor formal remarks:
- This is not a taxonomic study. Nevertheless, given the dynamics of views on lion taxonomy, I would consider it appropriate that the authors clearly state explicitly (in the context of current views on lion taxonomy) why (resp. with reference to which studies) they use the subspecies designation Panthera leo persica.
- I recommend, for taxonomic precision, eventual considering that when a scientific name of a species is first mentioned in the text, the author of the name should also be mentioned (lines 27, 30, 32, 33, etc.).
- Alaska and Yukon are part of North America (line 28)
- Unify the formal writing of references to multiple authors (sometimes it is without a punctuation comma before the year - e.g. Kittle et al. 2016, other times with a comma e.g. Jhala et al., 2019).
- Line 480 - second Acknowledgments - apparently it should be „abbreviations used“.
Author Response
The present study is a formally, methodically and factually well-crafted contribution to the knowledge of the conservation status of Asiatic lions and an important contribution to the practice and theory of their conservation.
- Thank you!
I strongly recommend its acceptance. I have no major reservations. Just a few minor formal remarks:
- This is not a taxonomic study. Nevertheless, given the dynamics of views on lion taxonomy, I would consider it appropriate that the authors clearly state explicitly (in the context of current views on lion taxonomy) why (resp. with reference to which studies) they use the subspecies designation Panthera leo persica.
- Now addressed to justify the subspecies designation we use in the second paragraph of the introduction and cite the relevant articles to address this. “It is also notable to mention that according to revised taxonomic Asiatic Lion is considered as a Asiatic subpopulation of P.l. leo [8], however, for conservation purposes, legal aspects (As P.l. persica is a mentioned under Schedule-1 with special protection from the Wildlife Protection Act 1972 of India [9]) and current usage in recent peer-reviewed publications [10,11], the use of sub-species name P. l. persica is continued in this study as well.”
- I recommend, for taxonomic precision, eventual considering that when a scientific name of a species is first mentioned in the text, the author of the name should also be mentioned (lines 27, 30, 32, 33, etc.).
- Now addressed and the original author of species taxonomic names have been mentioned.
- Alaska and Yukon are part of North America (line 28)
- Rephrased the sentence and changes have been applied.
- Unify the formal writing of references to multiple authors (sometimes it is without a punctuation comma before the year - e.g. Kittle et al. 2016, other times with a comma e.g. Jhala et al., 2019).
- ACS style has been applied as per Journal guidelines for references which eliminates the necessity of these changes.
- Line 480 - second Acknowledgments - apparently it should be „abbreviations used“.
- Now it has been corrected as 'Abbreviations'.
Reviewer 2 Report
Comments and Suggestions for Authors
My comments are included in the attached file

Comments on the Quality of English LanguagePlease reorganize the introductory part (as shown in the attached file) and rephrase the suggested paragraphs
Author Response
- Lines 50-51 – please provide more possible explanations why the lion population has increased by 50 137.32% and its geographic area has expanded by 354.55%, during the last three decades;
- Now provided based on findings of the two references cited therein.
- Line 57 – goes together with the new paragraph, and should be grouped together;
- Grouped as suggested.
- Line 65 - “the lack of genetic diversity can lead to inbreeding”, more correct I consider that it is vice versa – due to the inbreeding the genetic diversity resulted;
- Sentence rephrased as per the article to convey the correct meaning of the statement.
- Lines 71-74 – should be rephrased, because the conclusions are unclear. Please read carefully Shankaranarayanan et al. 1997 article and present its conclusions correctly
- Rephrased and conclusions rechecked and presented correctly.
- Lines 76-92 - suggestion to reorganize into two main comparative components the resulted researches – pro, on one hand and against on the second hand, because how it is organized now it can create confusions;
- As the coherence of the Introduction depends upon description of the Bottleneck events, this is has been rearranged to address this concern.
- Lines 102-106 – please rephrase this, and please be careful to the English used;
- Rephrased and English usage improved to convey meaningful explanation.
- Lines 125-149 – needs reorganization of the info, on males and females and after that regarding different categories for each sex (it is a little amalgamated);
- Addressed and wherever required the sex is mentioned.
- Line 164 – which is the total area of Gir Protected Area aka Gir 164 PA?
- The total areas are now mentioned in Methods, Study Area sub-section.
- Which is the total number of the lions in the area?
- Latest verified estimates mentioned in last paragraph of Introduction and appropriate reference cited.
- Please review the English and reorganize the introductory part.
- Addressed after incorporating all editor and reviewer suggested changes.